# Design and Implementation of 2D MIMO-Based Optical Camera Communication Using a Light-Emitting Diode Array for Long-Range Monitoring System

**DOI:** 10.3390/s21093023

**Published:** 2021-04-26

**Authors:** Huy Nguyen, Vanhoa Nguyen, Conghoan Nguyen, Van Bui, Yeongmin Jang

**Affiliations:** Department of Electronics Engineering, Kookmin University, Seoul 02707, Korea; ngochuy.hust@gmail.com (H.N.); vanhoahd95@gmail.com (V.N.); conghoan.tdhbka@gmail.com (C.N.); buivandut@gmail.com (V.B.)

**Keywords:** internet of things, multiple-input multiple-output, on–off keying, optical camera communication

## Abstract

Wireless technologies that use radio frequency (RF) waveforms are common in wireless communication systems, such as the mobile communication, satellite system, and Internet of Things (IoT) systems. It is more advantageous than wired communication because of the ease of installation. However, it can negatively impact human health if high frequencies are used to transmit data. Therefore, researchers are exploring the potential of optical wireless communication as an alternative, which uses the visible light bandwidth instead of RF waveforms. Three possibilities are being investigated: visible light communication, light fidelity, and optical camera communication. In this paper, we propose a multiple-input multiple-output modulation scheme using a light-emitting diode (LED) array, which is applicable to the IoT system, based on on–off keying modulation in the time domain. This scheme is compatible with the two popular types of camera in the market, rolling shutter cameras and global shutter cameras, as well as the closed-circuit television camera, which is used in factories, buildings, etc. Despite the small size of the LED array, implementing this scheme with 10 links in different positions at a communication distance of 20 m is possible for efficient performance (low error rate) by controlling the exposure time, shutter speed, focal length, channel coding and applying the matched filter.

## 1. Introduction

Currently, we encounter the Internet of Things (IoT) system frequently in our everyday lives, such as in smart homes, e-health systems, traffic monitoring systems, smart grids, and smart factories [1]. With the IoT system, devices can quickly join the IoT network through applied internet protocols [2]. The IoT system includes module devices, microcontroller units, plants, animals, and humans. It is designed to automatically connect people worldwide to the IoT network, which can result in a lack of human interaction. The IoT has played a significant role in the fourth industrial revolution; it helps people around the world to easily connect to one another. The wireless communication techniques upon which the IoT depends include radio frequency (RF) waveforms like LoRa, ZigBee, WiFi, and Sigfox. Furthermore, the negative effects that RF waveforms have on human health must be considered [3], especially in hospitals, schools, and other places where there are older people, children, and patients. Because of the harmful effect of RF systems, several researchers are assessing new suitable alternative technology. The use of visible light to transmit data, i.e., optical wireless communication (OWC), has emerged as a promising candidate to replace RF techniques. There are three possible techniques that can be used in OWC: visible light communication (VLC), light fidelity (LiFi), and optical camera communication (OCC). In LiFi and VLC, photodiodes use as the detectors that receive the intensity of light-emitting diodes (LEDs). Unlike LiFi and VLC, in OCC, image sensors serve as the detectors that receive data. Global shutter cameras and rolling shutter cameras, two popular camera types, are used in OCC systems [4]. Based on the type of camera, suitable OCC modulation schemes can be designed. The benefits of OWC compared with those of RF are presented below:RF waves are commonly used in communication and have ubiquitous advantages, but it has two significant drawbacks: its impact on human health; and the influence of electromagnetic interference (EMI) on system performance. On the contrary, visible light waves do not have any known impact on human health [5].It is highly important to consider safety and effectiveness when the channel model achieves line-of-sight transmission.The bandwidth of a visible light wave is more than 1000 times the bandwidth of Radiofrequency.

With these potentials, numerous companies have funded research to develop and improve OWC technologies. In IEEE 802.15.7-2011 standard [6], the OWC techniques wewe introduced with its complexity protocol in 2011 only depend on VLC mode. The IEEE 802.15.7-2018 [7] is published, which amend the previous standard. In IEEE 802.15.7-2018, the OWC technique is amended with the following four modes:The IEEE 802.15.7-2011 standard already provides detailed information about VLC modes [6].OCC: Modulation schemes can receive data from a variety of light sources using image sensors or cameras.High-speed LiFi: The data rate is significantly improved to achieve more than 1 Mbps at the physical layer using high-rate photodiode modulation schemes.LED identification: It mentioned communication schemes that use photodiodes to transmit data at a low data rate (below 1 Mbps).

Owing to innovative manufacturing techniques, LEDs have the potential to become next-generation lights due to several advantages, such as long life, efficient power consumption, and many different modes and sizes. Besides that, one characteristic that makes LEDs useful for OWC technologies is the fast switchability [7,8], which enables a high-rate OWC system.

RF systems currently have numerous applications, such as communication, monitoring, and radar systems. Notwithstanding these applications, RF generates electromagnetic interference, affecting human health and brand function [9,10]. OWC techniques have gained attention because they are EMI-free and thus will likely replace the RF technique [11] in the future. With VLC/LiFi, photodiodes receive LEDs’ intensity based on the switching ON/OFF statuses from LEDs [12]. Refs [13,14] implemented an ultra-high-speed pulse density modulation with extremely effective spectral properties using photodiodes. The multiple-input multiple-output (MIMO) technique is applied to transmit high-speed data with ultra-high-speed multi-channels for the VLC and LiFi system [15,16]. Because a photodiode is used as the detector, the VLC and LiFi techniques have some disadvantages: the communication distance is short, making them compatible with indoor applications only. In an outdoor environment, the channel distortion obstructs the LED signal receiving photodiodes. However, the OCC system, which deploys an image sensor as the receiver, then the OCC can operate at a distance of up to 200 m [17]. Nguyen et al. [18] presented the effect of image sensor types on OCC performance. When we use a global shutter camera for OCC system, the camera frame rate and packet rate must be considered carefully based on Nyquist’s law. When we use a rolling shutter camera, the sampling rate should be considered based on the camera frame rate and the rolling shutter speed. The signal-to-noise ratio (SNR) should be considered to transmit data with long communication distance besides the focal length of the camera and exposure time. LiFi technology can now achieve a maximum communication distance of 10 m using a photodiode lens [19].

With OCC techniques, the MIMO technique can operate efficiently, resolving simultaneous links with multiple-light sources and the camera. However, with photodiodes, the MIMO technique is not feasible for multiple connections. Region of interest (RoI) signaling is a popular algorithm that can be applied to detect multi-light sources in an OCC system. The authors in [20] proposed the camera on–off keying (OOK) with a high data rate based on the rolling shutter effect. Still, this scheme has some drawbacks, including a limited communication distance and low performance. In [21], the authors proposed combining the orthogonal frequency-division multiplexing (OFDM) waveform and rolling shutter effect (rolling-OFDM) for a high-data-rate OCC system. Because intensity modulation/direct detection (IM/DD) modulation is deployed, the rolling-OFDM scheme requires suitable LED materials. A 2D-OFDM scheme based on the screen code with a high data rate is proposed by Nguyen et al. [22]. However, its drawbacks include the large transmitter size, the short communication distance, and the expensive transmitter.

In [23], the authors proposed a color intensity modulation-MIMO scheme to transmit data using a global shutter camera, which has a frame rate of 330 frames per second. Because it uses color intensity modulation, a maximum communication distance is just 1.4 m with high BER value (BER of $10^{-1}$). This scheme is just compatible with the global shutter camera, which is expensive and not widely used. Moreover, using colors to transmit data has some drawbacks compared with OOK schemes, such as the limited communication distance and high bit error rate. In [24], the authors proposed a rolling-shutter-based MIMO scheme with an LED matrix. This scheme can resolve the flicker effect, but there are some disadvantages: a short communication distance (1.4 m) and non-rotation support (such support is essential when using 2D-code for the OCC technique for IoT systems). In this paper, we will propose a monitoring system using an LED-array-based OCC system. The MIMO technique, RoI signaling, and matched filter are deployed to improve the performance of the OCC system. The rolling shutter camera and global shutter camera can easily to implement with the proposed scheme in the OCC system, especially in closed-circuit television (CCTV), which is commonly used; thus, this scheme can be applied to an environment monitoring system that uses CCTV as a receiver. The proposed approach can achieve a long communication distance of 20 m with a low bit error rate.

The remainder of this paper is divided into four sections. Section 2 mainly focuses on the contributions of the proposed scheme. Section 3 provides detailed information about the architecture of the proposed system. In Section 4, we show the implementation results to demonstrate our contributions to the proposed scheme. Finally, Section 5 details our conclusions.

## 2. Our Contributions

In this paper, we propose OCC for an IoT-system-based 2D MIMO technique using an LED array compatible with almost every camera in the market. Our scheme has several advantages, which are highlighted as follows:Support for most commercial cameras: Our proposed scheme can operate with rolling shutter cameras and global shutter cameras because of its controllable exposure time. Furthermore, due to its RoI detection algorithm, this scheme can easily be deployed with CCTV, widely used, making it convenient.Rotation support: This scheme can allows 360-degree rotation because the matrix uses bit-based rotation. By reasonably arranging four corners of the LED array, receivers can easily detect and support 360 degrees of rotation. This is important for IoT systems because when rotation support is provided, cameras, especially CCTV cameras, can easily receive data from every angle in a real-world setting. The rotation support algorithm is presented in Section 4.Lower bit error rate (BER) performance: This scheme applies the matched filter, so it performs better than the conventional zero-crossing filter, determining the on/off thresholds in each image.Frame rate variation support: Frame rate variation occurs when the packet is missing in the decoding process at the receiver side, which is an important consideration when designing an OCC system. Most people think that the frame rate described on the label of cameras is constantly based on the camera’s specifications (e.g., 30 and 60 fps). However, the frame rate varies between the transmitter and receiver sides, making it difficult to synchronize. With the sequence number (SN) in each sub-packet, the receiver side can easily check whether the data sub-packet has been dropped.Data merger algorithm: In our proposed scheme, we insert the SN in each sub-packet, which represents the sequence number of sub-packet. Depending on the length of a data packet, we can select the SN length to increase system performance.Detection of missing packets: To merge packages from two consecutive images, we propose the SN, which is inserted into each package. Using SN, we can easily detect the missing packets by comparing the SNs in two consecutive images if the SN length is sufficient.

## 3. System Architecture

The main concept of an OCC system is to control the intensity by which optical signals transmit and receive data; then, with a promising modulation scheme, the system communication performance can be enhanced. OOK is a well-known modulation scheme and has the simplest amplitude-shift keying modulation. It uses two statuses, “on” and ”off,” to transmit data, represented by bit “1” and bit “0,” respectively. In this paper, we propose the details of an LED-array-based scheme, which is applied for an IoT-system-based OCC technique. By locating each LED in an LED array based on our proposed scheme’s spatial frame format, the receiver side can easily detect and decode data based on RoI algorithms. Figure 1 presents the architecture of the 2D OOK–MIMO transceiver. The operation of the blocks is discussed in Section 3.

### 3.1. Channel Coding

Channel coding is a significant part of numerous digital communication systems. It is also well known as a forward error control (FEC) scheme that can detect and reduce the bit errors in the digital system. Channel coding can be used in both the transmitter and receiver to increase the reliability of the system. Channel coding is used to encode on the transmitter side by adding extra bits to raw data before modulation. Otherwise, channel coding is used to decode on the receiver side. By applying channel coding, errors in the received side can be easily detected and corrected. In our experiment, the suitable channel coding was selected based on the size of the LED matrix.

### 3.2. Add Start of Frame and SN

The preamble is the first part of the symbol frame. It is applied to synchronize and detect the start of frame and helps the receiver side to decode data easily. If the preamble part is redundant (i.e., if it does not carry the desired data), then the length of the preamble should be carefully considered. In this paper, we proposed a preamble length of 6 bits (SF = “011100”). With the preamble, we can easily detect the start of frame in each image.

The sequence number (SN), which helps the receiver side combat the frame rate variation effect, is inserted into each packet after the preamble. It helps to remove the redundant packets in case of oversampling. Next, the large package will be merged from adjacent images. The SN also enables the cameras in the receiver sides to detect the missing packet in case of undersampling. The advantages of the SN are presented in Section 4.

### 3.3. LED Detection

RoI algorithms are already well known because of their high performance in object detection. Both object and feature-based detection approaches are used in most RoI algorithms [25].

The information regarding the shapes and structures in the object-based method is used for detection. The feature-based method uses the optical features of the collected pixel to apply RoI algorithms. The main problem with this approach is that if all pixels do not have a sustainable feature, this method is inefficient for detection. Consequently, the object-based approach is better than the feature-based approach in terms of RoIs.

In this paper, an RoI algorithm is used to detect the positing anchor at four corners of an LED array, which is clearly presented in Section 4. The four outermost corners of the LED array are always in the “on” status, which helps the receiver side easily detect via a RoI algorithm. From the coordinates of the four outermost corners in each image, the coordinates of another LED can be easily detected and decoded. In cases with multi-users and 360-degree rotation support, we applied artificial intelligence (AI)-assisted detection to increase system performance.

### 3.4. Match Filter

In Section 4, we will show the relationship between SNR and communication. With the higher communication distance, the SNR values will be smaller. Therefore, the receiver side will be challenging to define the threshold between ON and OFF values. We should find another technology to replace the zeros crossing technology to increase the communication distance.

The matched filter, known as a linear filter technology, maximizes the SNR value with additive random noise in the communication system. It is widely applied in wireless communication systems to optimize the SNR values at a low cost. The 802.15.7-2018 standard [7] recommends using the matched filter in an OCC system. This paper proposes using the matched filter to optimize the SNR value and increase the communication distance. As shown in Figure 2, the signals from 48 LEDs were combined based on the intensity of 48 areas, which were determined from four positioning anchor corners. By generating known patterns such as those in Figure 3, the received signals from two adjacent LEDs will multiply with the 1 pulse pattern and the 0 pulse pattern via the convolution algorithm. Therefore, we can easy to define the signal value (1 or 0).

## 4. Implementation

### 4.1. Pixel Eb/N0 Computation and Noise Modeling

In CCD/CMOS, the pixel noise in cameras can be approximately modeled using Equation (1), according to [23].
(1)$$n\sim N(0,\delta {\left(s\right)}^{2})$$
where *s* represents the value of a pixel, $\delta^{2}(s)=s\times a\times \alpha +\beta$, *a* is the mark and space amplitude, and α, β specify the fitting factors obtained in the experiments. In our implementation, we used model-fitting coefficients in our system, which were able to predict experimentally [26]. The pixel $E_{b}/N_{0}$ in the receiver side was estimated using Equation (2) [26], with the assumption that one symbol contains one bit.
(2)$$Pixel\frac{E_{b}}{N_{0}}=\frac{E\left[s^{2}\right]}{E\left[n^{2}\right]}\approx \frac{a^{2}\times \Delta }{a\times \alpha \times \Delta +\beta }$$
where $E_{b}$ is the bit energy, $N_{0}$ represents the noise density, *s* is the value of a pixel, $\Delta =T_{\exp osure}/T_{bit}$ is the ratio of camera exposure time to the bit interval, and α, β are the fit parameters.

Because of the difference between camera types on the market, each camera should indicate its evaluation. For instance, a theoretical estimation of pixel SNR is illustrated in Figure 4 as a nonlinear curve that used Equation (2). The shutter speed of a rolling shutter camera is estimated to be 10 kHz.

### 4.2. Experimental SNR Measurement

The SNR values were measured to determine the relationship between the communication distance, SNR value, and exposure time. By controlling the camera’s exposure time, the desired communication distance and BER performance can be achieved. In the experimental SNR measurement, we deployed several measures with communication distances of 5–20 m.

We implemented the proposed system using a Point Grey camera with rolling shutter mode on the receiver side and a LED matrix (5-V, LED matrix 8 × 8, 8 × 8 cm^2^ rectangle) on the transmitter side shown in Figure 5. Moreover, we determined the performance of the proposed system by controlling the various values of camera exposure time (100–750 µs) at different communication distances.

The experiment was conducted at distances of 5, 10, 15, and 20 m with the LED statuses set to both “ON” and “OFF to measure SNR values.” With the “ON” status, the SNR was calculated as the signal power of the LED and is shown as a red line with different communication distances. However, the SNR is considered background noise when the LED is in the “OFF” status. In this case, the SNR is displayed as a blue line with a different distance. In both transmission statuses, the SNR (dB) was calculated based on the experiment using the following:(3)$${\mathrm{SNR}}_{\mathrm{dB}}=20\times \log(\frac{\sqrt{\frac{1}{n}\sum_{i=0}^{n-1}{\left|A_{i}\right|}^{2}}}{\sqrt{\frac{1}{n}\sum_{i=0}^{n-1}{\left|B_{i}\right|}^{2}}})$$

In Equation (3), *A* is the measurement of the LED’s signal power, *B* is the measurement of the background noise, and n is the number of measured samples. For example, Figure 6, Figure 7, Figure 8 and Figure 9 shown that: the LED’s signal power is displayed in the pink line, and the background noise is displayed in the green color line with different communication distances. The received pixel amplitude value varies inversely with the distance, i.e., it is smaller at longer distances. The SNR calculation is also affected by the shuttle speed and exposure time, as illustrated in Figure 10. In this scenario, the operation of an image sensor is similar to that of a low-pass filter, which smooths the high-tone signal with an increase in the exposure time. Similarly, the communication bandwidth decreases, leading to the smaller noise power spread over the bandwidth when the exposure time increases. Based on Equation (3), the SNR will increase with a longer exposure time because of decreased noise power. We also examined the relationship between the SNR and communication distance (Figure 10). As the distance increases, the SNR value decreases because of the smaller light sources received by the image sensors.

### 4.3. BER Estimation for Optical OOK Modulation

The electrical amplitude of the received signal in the receiver side of the communication system is calculated as follows:(4)$$r(t)=I(t)+\sum_{i=-\infty }^{+\infty }I(t)\times a_{i}\times g(t-iT_{symbol})+n(t)$$
where $a_{i}$ is the level of the ith OOK symbol and $a_{i}$∈ {0,1}. We assumed that the probabilities of bit 0 and 1 are $P_{0}$ and $P_{1}$, respectively, g(t) is rectangular function; and $T_{symbol}$ is the symbol duration. Because of only additive white Gaussian noise (AWGN) in the channel model, the BER can be presented [26] in Equation (5).
(5)$$P_{e}=\frac{1}{2}\times erfc(\sqrt{\frac{E_{b}}{2\times \sigma_{n}^{2}}})$$

The received signal on the receiver side for bits 0 and 1 in the AWGN channel is as follows:
(6)$$r(t)=\left\{ \begin{array}{l} n(t),a_{i}=0\\ 2\times I(t)+n(t),a_{i}=1 \end{array}\right.$$

Figure 11 presents the estimation results with different SNR values for the matched filter and zero-crossing filter, which is compared with the theoretical bit error probability of OOK modulation. The performance of our proposed scheme is better than the zero-crossing filter. Based on the information given in Figure 10 and Figure 11, we can achieve a BER of $10^{-4}$ at an SNR of 11 dB, with an exposure time of 100 μs, and a communication distance of 20 m if the matched filter is used. We achieved a BER of $10^{-4}$ at an SNR of 16 dB, which corresponds to a communication distance of 10 m with a 100-μs exposure time. From this, it is evident that the performance of the OCC system can be improved by applying a matched filter. The exposure time can be adjusted to increase the SNR, but the exposure time needs to be carefully considered because a long exposure time can decrease the system bandwidth.

### 4.4. Proposed Scheme

Figure 12 displays the spatial frame format of our proposed scheme in an 8 × 8 LED matrix. The four outermost corners of the LED array were used as anchors for corner detection. From the coordinates of these corners, all the LED coordinates can be easily calculated on the basis of perspective transformation. We used 40 LEDs for data transmission in this LED matrix, as shown in Figure 12. In this scheme, we proposed four-position anchor parts in four corners (Figure 12) with a particular positioning of anchor 3, which supports rotation on the receiver side. With the Zeros Crossing technique, 16 LEDs used as position anchors served as the training signal part, which helped the camera to determine the threshold between the “ON” and “OFF” statuses of all LEDs. The preamble, which helps the receiver side to detect the start of frame easily, was inserted into each frame. Matched filter technique is applied with the line coding to optimize the SNR value. Thus, the communication distance is improved. The artificial intelligence (AI) algorithms were applied to help detecting multiple LED array with mobility effects to increase system performance. The SN was proposed to reduce the effect of frame rate variation in the camera, and the serial number of the packet was used as an SN. In the implementation, based on the relationship between the packet rate on the transmitter side and the camera frame rate on the receiver side, the SN length was accordingly considered and controlled. If the camera’s frame rate on the receiver side is smaller than the packet rate on the transmitter side, undersampling occurs in the OCC system. In contrast, if the camera frame rate on the receiver side is higher than the packet rate on the transmitter side, oversampling occurs. The packet rate on the transmitter side is defined as the number of packets transmitted in a single period (e.g., 30 packets per second). The packets can be identified by their SNs, which contain the serial information of each packet. Because of the effect of camera frame rate variation, the SN proved to be useful for decoding data because it enabled the camera to verify the newly arrived packet in the oversampling case and detect the losted packet in the undersampling case. Nguyen et al. [20] used 1-bit Ab for the oversampling case and 2-bits Ab for the undersampling case. This created confusion in the decoding process on the receiver side, which limited the number of missing payloads it was able to detect. Our scheme proposed using n bits of SN instead of Ab bits. The length of the SN can be modified depending on the implementation and the camera parameters. If conditions are not ideal or the camera is not of high quality, the number of missing packets is high, and the SN length is larger. However, the length of SN reduces with the quality of the camera and thus increases system performance.

#### 4.4.1. Oversampling

When the frame rate of the received camera is at least twice as large as the transmitter light source, oversampling occurs. On the receiver side, each package is captured at least twice and is duplicated, leading to confusion in the packet merger function; therefore, the SN is added to each data sub-packet (DS) to have DSs with the same payload and SN. Based on the SN, the receiver camera can select compatible DSs and remove redundant DSs caused by oversampling. The receiver camera will remove the duplicated packet and choose the packet with an increasing SN (n, n + 1, n + 2) to merge.

#### 4.4.2. Undersampling

When the frame rate of the received camera is lower than the transmitter light source, undersampling occurs. In this case, the DS is missed, confusing the packet merge function. By applying an eligible SN to each packet, all the missing payload can be recovered easily. The SN value of n is represented by the DS received from the packet n. Similarly, the SN value of n + 1 is represented by the DS received from the packet n + 1. By comparing two continuous SN values, the proposed system can verify that the packet n is dropped, and the payload could be missed if the actual DS possesses an SN value of n + 2. Moreover, the length of the SN is considered to calculate the number of missed payloads. For instance, if the SN length is 4 bits, the system can detect a maximum of seven missed payloads. If the system detects the two non-adjacent of the SN value within two adjacent payloads (e.g., n and n + 2), the missed payload n + 1 is found.

#### 4.4.3. Maximum Communication Distance

The proposed scheme can communicate with multiple users with only a single camera using RoI to detect the source of LEDs. However, the number of users is limited because of the difference in the focal length, transmit distance, and distance between two consecutive LEDs. Figure 13 shows this relationship.

The relationship between the focal length and work distance [27] is:(7)$$\frac{d\left[m\right]}{D\_\mathrm{LEDs}\left[m\right]}=\frac{f\left[\mathrm{mm}\right]}{d\_\mathrm{LEDs}\left[\mathrm{mm}\right]}$$
where the focal length of the camera and the work distance is represented by *f* and *d*, respectively. D_LEDs is the distance between two LEDs in a realistic experiment, and d_LEDs is the distance between two LEDs in the image. Equation (7) can be used to calculate the minimum distance between two LEDs. By applying the Nyquist theorem, RoIs can detect the minimum distance between two LEDs in two pixels. The value of the minimum distance between two LEDs is calculated as follows:(8)$$D_{\mathrm{LEDs}\left[m\right]}=2\;\times \;\frac{d\left[m\right]}{f\left[\mathrm{mm}\right]}\;\times \;\frac{h\_sensor\_image\left[m\right]}{N\_pixel\_image\_row}D\_LEDs[m]=2\times \frac{d[m]}{f[mm]}\times \frac{h\_sensor\_image[mm]}{N\_pixel\_image\_row}$$
where h_sensor_image and N_pixel_image_ are the height of the sensor image in millimeters and the number of pixel rows in the image, respectively. The theoretical distance is simplified by *d* [m] since the work distance also depends on the light power. The interference occurs when the power of the transmitter is smaller than that of the noise lights. Due to this reason, we can define the maximum number of users based on the monitoring environs.

#### 4.4.4. Implementation Results

In this paper, we deployed the proposed scheme several times with various cameras (a webcam, CCTV camera, tablet, Point Gray rolling shutter camera, etc.) to analyze the effect of camera frame variation. The asynchronous process includes asynchronous decoding, the merging data technique, and the detection of missing parts. The length of the SN is considered carefully based on the integrated operation. Figure 14 presents the quantized intensity profiles of captured images with different communication distances and degree perspective viewing angles. Figure 15 displays the setup of the proposed implementation, and Figure 16 shows the experimental results obtained by a tablet rear camera. The implementation results are presented in Table 1.

Figure 10 shows the relationship between communication distances, exposure times, and the intensity of the LED in the images (or SNR value). The exposure time can be increased to increase the communication distance, but as mentioned above, the exposure time should be controlled carefully. The image sensor operates as a low-pas filter. Then, if the exposure time is increased, the bandwidth of the OCC system will decrease due to the increasing power of the background noise.

Figure 17 shows the results of using a Point Grey rolling shutter camera at different distances and exposure times. The BER is measured with varying times of exposure, with the same communication distance and noise condition. The communication bandwidth has a vital relationship with the exposure time; thus, both exposure time and the trade-off between exposure time and signal noise must be considered. To reduce the bit error rate or increase the communication distance and thus increase system performance, channel coding should be used. However, to calculate the working communication distance, Equations (7) and (8) should be used. The parameters of the proposed scheme are presented in Table 1. The summary of the comparison of MIMO-OCC methods will be presented in Table 2 to highlight the proposed scheme. We implement this scheme with 8 × 8 LED matrix at the communication distance of 2 m, which can be visually assessed by the Appendix A.

## 5. Conclusions

In this paper, we proposed a monitoring system based on the 2D OOK–MIMO scheme for the OCC technique. By using an LED matrix based on a special spatial frame format, the proposed scheme facilitates the rotation effect, which is essential with two-dimensional (2D) code. The SN was inserted into each packet, which helps in supporting the frame rate variation effect and recovering large data packets from different images. The SNR measurement results were obtained using various communication distances and exposure times to verify the relationship between three parameters: communication distance, exposure time, and SNR value. The SNR increases with exposure time, but the system bandwidth decreases, so controlling the exposure time should be carefully considered. The sub-packet was merged to create a large packet, which was then transmitted via a signal by adding the SN to overcome oversampling and undersampling. With suitable SN length, the lost, redundant packet can be detected based on the non-sequential nature of SNs, leading to an improved BER. However, the SN does not carry data and decreases the data rate, leading to the trade-off with the BER. Finally, the different distances were considered to evaluate the bit error rate of the proposed scheme compared with other similar systems.

## Figures and Tables

**Figure 1 sensors-21-03023-f001:**
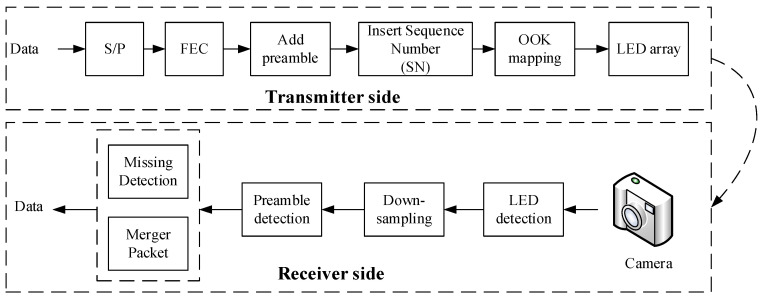
Reference architecture of OCC for an IoT-system-based 2D OOK–MIMO technique using an LED array.

**Figure 2 sensors-21-03023-f002:**
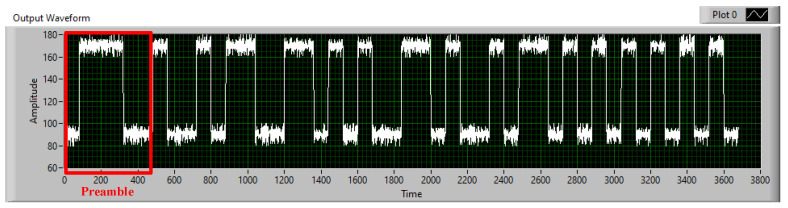
An experimental results of 2D-MIMO scheme from 48 LEDs (without 16 corner LEDs) in 8 × 8 matrix.

**Figure 3 sensors-21-03023-f003:**
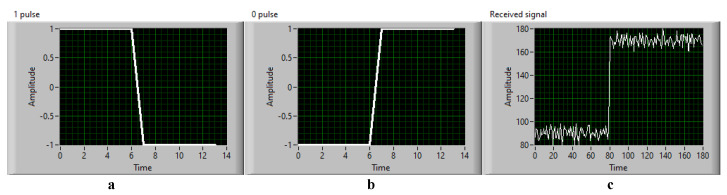
Manchester code patterns and the received signal from two adjacent LEDs (**a**) 1 impulse, (**b**) 0 impulse, (**c**) the received signal.

**Figure 4 sensors-21-03023-f004:**
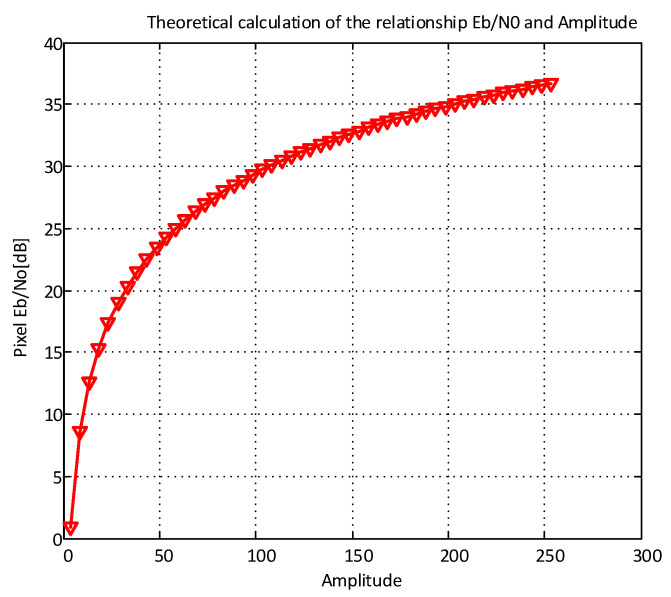
The theoretical relationship between Pixel Eb/No and image amplitude.

**Figure 5 sensors-21-03023-f005:**
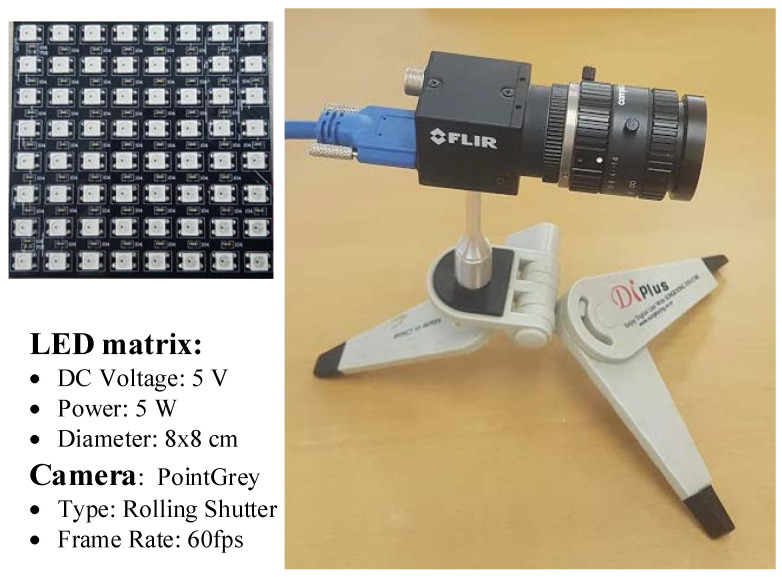
SNR measurement devices.

**Figure 6 sensors-21-03023-f006:**
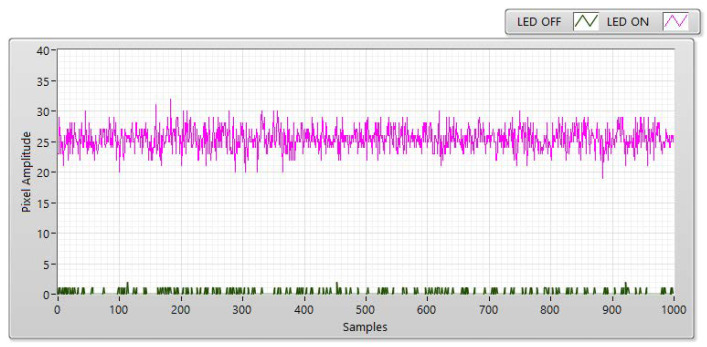
SNR measurement with a Point Grey rolling shutter camera at a communication distance of 5 m with 100 µs.

**Figure 7 sensors-21-03023-f007:**
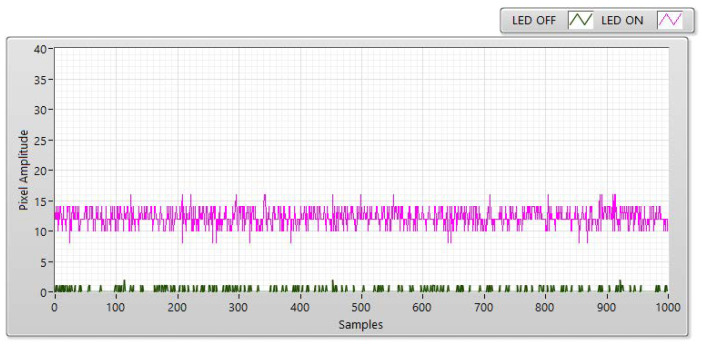
SNR measurement with a Point Grey rolling shutter camera at a communication distance of 10 m with 100 µs.

**Figure 8 sensors-21-03023-f008:**
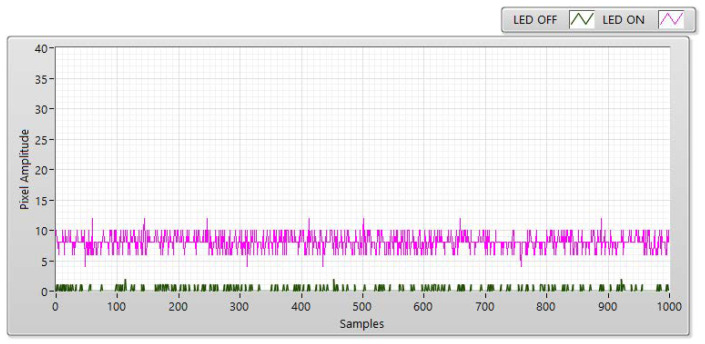
SNR measurement with a Point Grey rolling shutter camera at a communication distance of 15 m with 100 µs.

**Figure 9 sensors-21-03023-f009:**
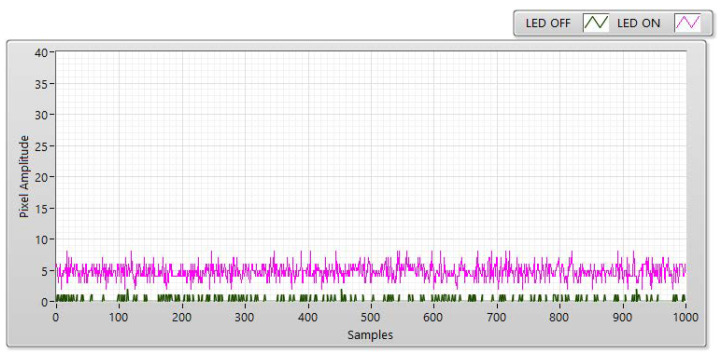
SNR measurement with a Point Grey rolling shutter camera at a communication distance of 20 m with 100 µs.

**Figure 10 sensors-21-03023-f010:**
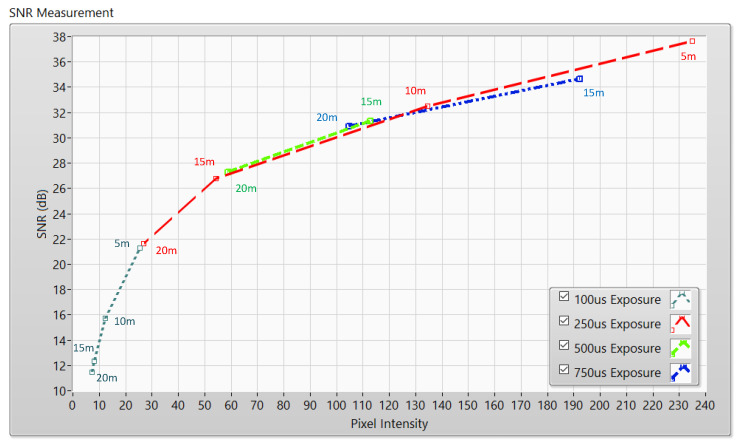
SNR measurement with different communication distances.

**Figure 11 sensors-21-03023-f011:**
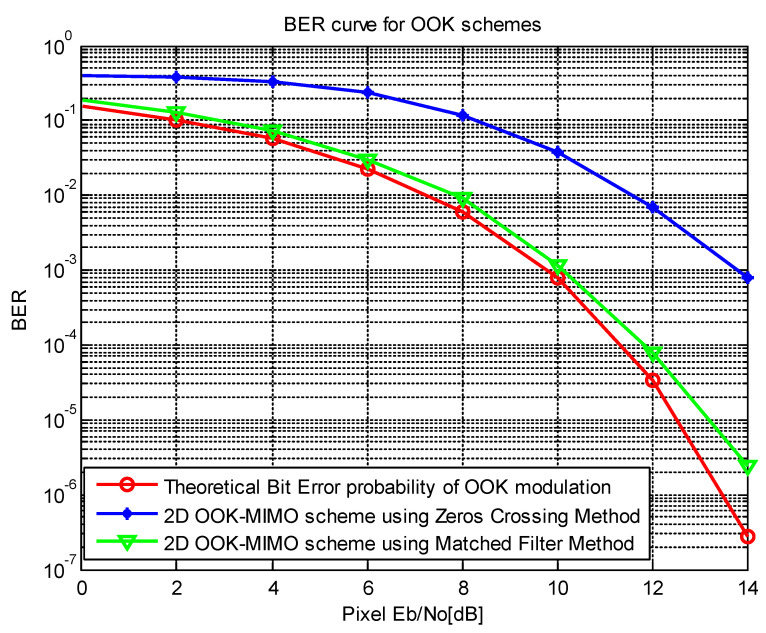
Bit error rate curve for optical OOK modulation schemes.

**Figure 12 sensors-21-03023-f012:**
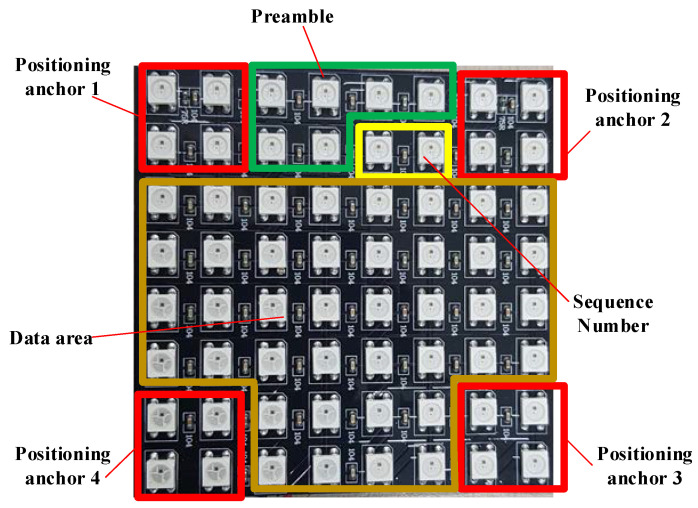
The spatial frame format in an LED array.

**Figure 13 sensors-21-03023-f013:**
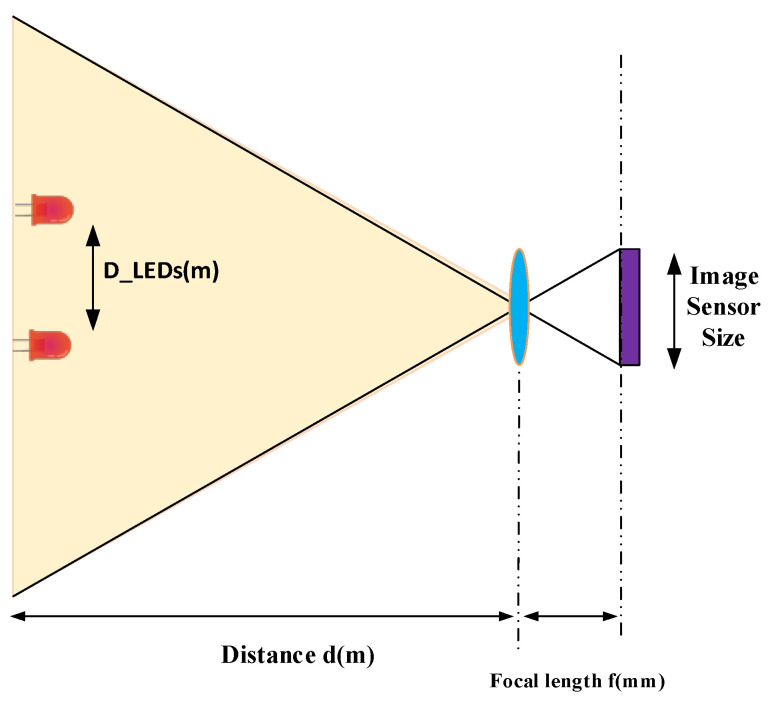
Relationship between the camera focal length and communication distance.

**Figure 14 sensors-21-03023-f014:**
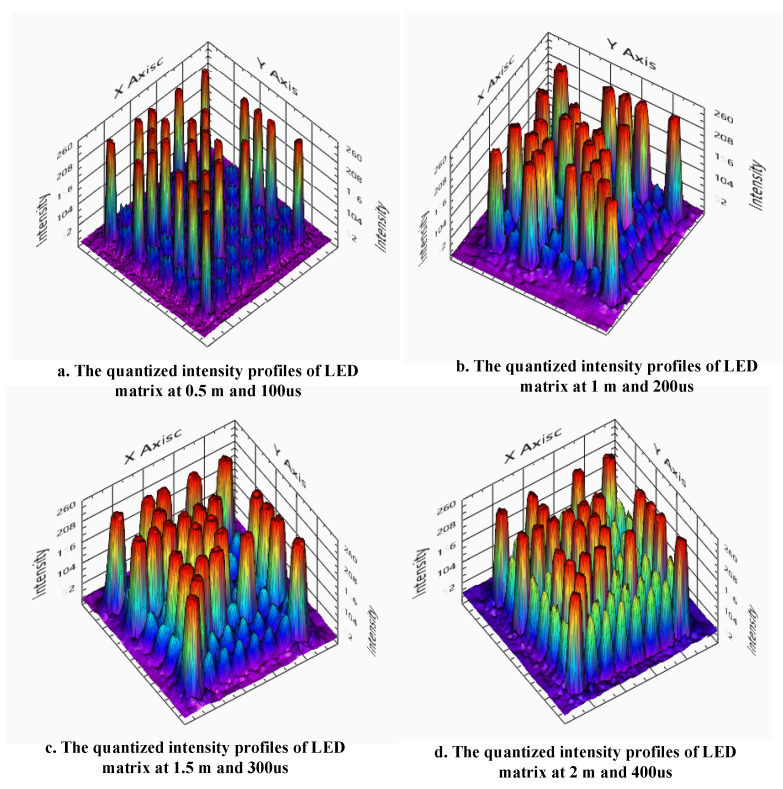
The quantized intensity profile of the LED matrix at (**a**) 0.5, (**b**) 1, (**c**) 1.5, and (**d**) 2 m with different exposure times.

**Figure 15 sensors-21-03023-f015:**
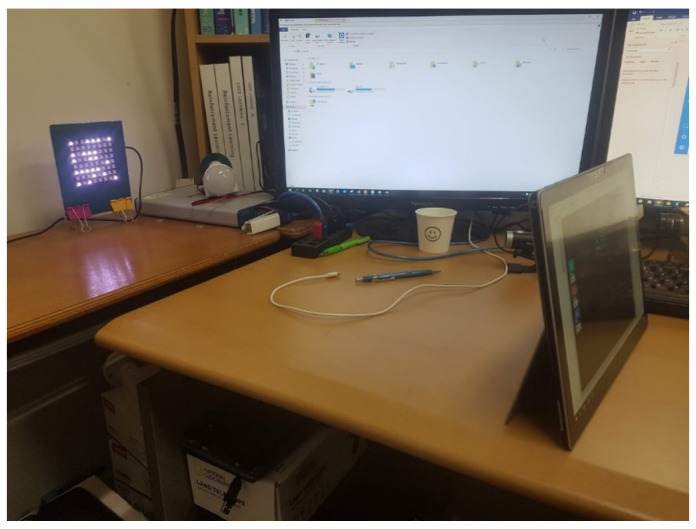
Setup scenario of proposed scheme with LED matrix and tablet.

**Figure 16 sensors-21-03023-f016:**
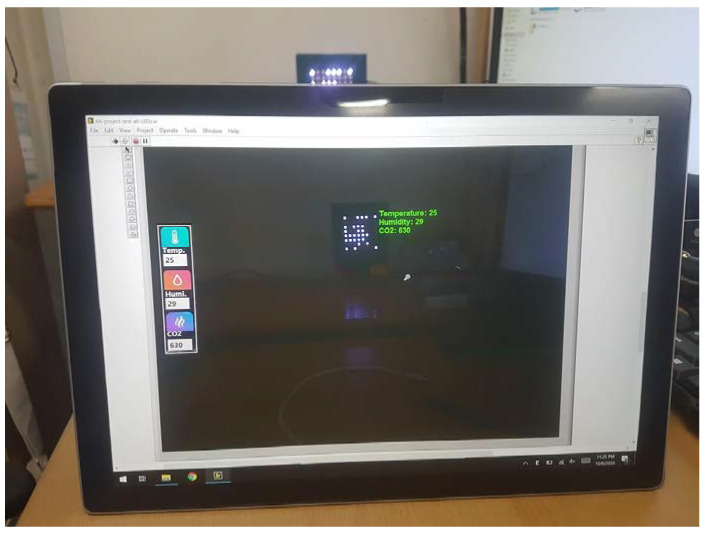
Rx interface with the rear camera of tablet.

**Figure 17 sensors-21-03023-f017:**
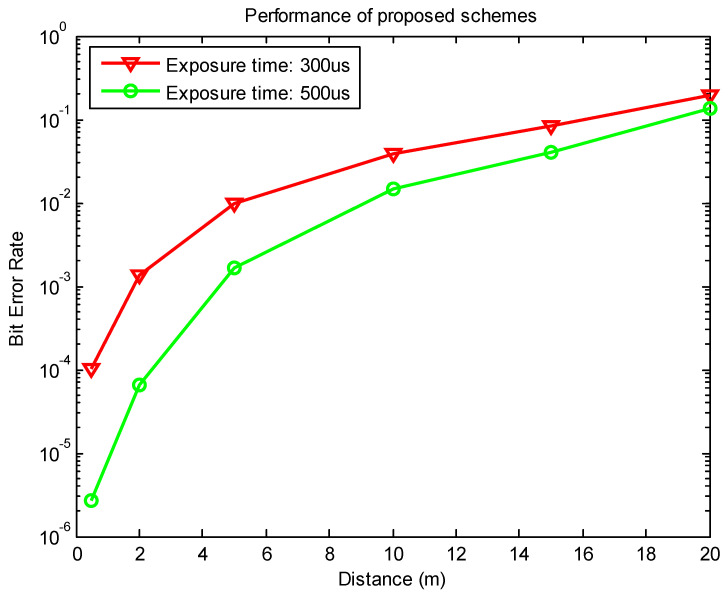
BER performance of the proposed scheme for an 8 × 8 LED matrix with different exposure times.

**Table 1 sensors-21-03023-t001:** Characteristic parameters of the proposed scheme.

**Transmitter Side**		
LED matrix size	8 × 8	16 × 16
The number of LEDs	64	256
FEC	CC(3/4)	
Packet rate	30 packet/s	
**Receiver Side**		
Camera	Point Grey rolling shutter camera	
Camera Frame Rate	60 fps	
**Throughput**		
Uncode bit rate	1.920 kbps	7.680 kbps
Code bit rate	1.440 kbps	5.760 kbps

**Table 2 sensors-21-03023-t002:** Summary of comparison of MIMO-OCC methods.

	Color Intensity Modulation-MIMO Scheme [23]	Rolling-Shutter-Based MIMO Scheme [24]	Proposed Scheme
Modulation	Color intensity	On-Off keying	On-Off keying
Type of camera	Just global shutter camera	Just rolling shutter camera	Both of cameras
Distance	Short distance (1.4 m)	Short distance (1.4 m)	Long distance (20 m)
Number of links	Single link	Single links	Multiple links
Matched filter	No	No	Yes
Bit error rate (1.4 m)	$>10^{-2}$	Not mentioned	$<10^{-4}$

## Data Availability

Not applicable.

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
