# Peer review of "Design and Implementation of 2D MIMO-Based Optical Camera Communication Using a Light-Emitting Diode Array for Long-Range Monitoring System"

_sensors, 2021, doi:10.3390/s21093023_

Round 1
Reviewer 1 Report
- The SNR is a very important for long communication distance, how can improve SNR? How about the maximum SNR can be obtain by using this method?
- The author should give a summary ofthe comparison with other methods.
- English must be improved especially on page 2.
Reviewer 2 Report
The authors investigate three possibilities : visible light communication, light fidelity, and optical camera communication.
They propose a multiple-input multiple-output modulation scheme using a light-emitting diode (LED) array,
which is applicable to the IoT system, based on on–off keying modulation in the time domain.
This scheme is compatible with the two popular types of camera in the market, rolling shutter cameras and
global shutter cameras, as well as the closed-circuit television camera, which is used in factories,
buildings, etc. Despite the small size of the LED array, implementing this scheme with 10 links in
different positions at a communication distance of 20 m is possible for efficient performance (low error rate)
by controlling the exposure time, shutter speed, focal length, channel coding and applying the matched filter.
The paper is well organized:
contains a list of contributions, system architecture, measurements and comparison with theoretical estimate.
This paper is exceptional and should be published in the journal with minor revisions.
1. Mention Figure 4-7 from line number 273 to 285.
2. Please make a comparison table of different approaches and
how your approach is better than other approaches.
Reviewer 3 Report
Interesting proposal and of great application. I enjoyed reading it.
Author Response
We are grateful for the consideration of the review of our work.
Reviewer 4 Report
In this paper the authors present a monitoring system using a 2D OOK–MIMO scheme in an optical camera communication system to improve the preview OCC schemes in terms of rotation effect, support for the variation of the frame rate and recovery of large data packets from different images.
I only have a few minor revisions:
- on line 81 of page 2 the term "switching off/off" is incorrect. Please correct;
- in the section 3 the blocks of the comunication architecture are described, but the organization of the subsection is confused. Reorganize the subsections following the order of execution of the operations;
- the equation (2) is not present in the reference [23]. Insert the correct reference;
- In the equations use the moltiplication dot instead of the dot.
